# Freestanding Metal Nanomembranes and Nanowires by Template Transfer with a Soluble Adhesive

**DOI:** 10.3390/nano12223988

**Published:** 2022-11-12

**Authors:** Peipei Jia, Xinzhong Wang, Xiaobing Cai, Qiuquan Guo, Dongxing Zhang, Yong Sun, Jun Yang

**Affiliations:** 1Shenzhen Institute for Advanced Study, University of Electronic Science and Technology of China, Shenzhen 518110, China; 2Shenzhen Institute of Information Technology, Shenzhen 518172, China; 3School of Aerospace Engineering, Xi’an Jiaotong University, Xi’an 710049, China

**Keywords:** template transfer, freestanding, nanomembrane, nanowire, metal-assisted chemical etching

## Abstract

The fabrication of nanostructures usually involves chemical processes that have in certain steps. Especially, it is necessary to use the chemical etching method to release the as-patterned structures from the substrate in most of the transfer techniques. Here, a novel scheme of template transfer as developed for the fabrication of freestanding Au nanomembranes and nanowires by using a soluble PVP adhesive. The nanomembranes feature the periodic nanohole arrays with high uniformity. Without the substrates, these plasmonic nanohole arrays show symmetric and antisymmetric resonance modes with bright and dark spectral features, respectively, in transmission. Through the spectral analysis for reflection, we have disclosed that the usual dark mode in transmission is not really dark, but it reveals a distinct feature in reflection. Two coupling modes present distinct spectral features in transmission and reflection due to their different loss channels. To show their versatility, the freestanding nanomembranes were also employed as secondary templates to form Si nanowire arrays by the metal-assisted chemical etching method.

## 1. Introduction

Freestanding nanomembranes have been of research interest for several decades since they combine nanoscale thickness and features with macroscopic lateral dimensions at the same time [1]. Several cutting-edge freestanding ultrathin membranes are made of inorganic materials (i.e., silicon [2,3], metals [4], nanoparticles [5], graphene [6,7], PDMS [8]), organic materials (i.e., epoxy resin [9]) and hybrid composites [10,11,12]. These freestanding membranes are emerging as critical elements in various sensing devices, such as in mechanical, chemical and thermal sensors [6,8,11]. For plasmonic sensing, freestanding metal membranes as miniature passive plasmonic sensors are highly desirable as they can be attached to unconventional substrates which are incompatible to conventional fabrication methods. In addition, the freestanding nanomembranes enable the access of the target into their structures without substrate blocking occurring [13,14]. For example, the freestanding nanohole array membranes can work as sample supports in single-particle cryoelectron microscopy with sub-1 Å specimen movement [15].

To date, several techniques have been used for the fabrication of freestanding nanoscale membranes, including spin-coating [16], layer-by-layer assembly [17,18,19,20,21], and monolayer self-assembled [22,23]. However, these approaches all involve chemical process. For example, we have developed a template transfer technique for the fabrication of freestanding Au nanomembranes with nanoholes [24]. This simple replication-releasing scheme has the advantage of template reuse. However, the previous method has two obvious drawbacks. Firstly, it is not applicable to nanowires. Once they are released from the template without any supports, the nanowires would become tangled like noodles do. Second, a sacrificial-layer etchant was used to release the membranes from the template. This may not only lead to a large volume of chemical waste, but also, it contaminate the final membranes. Likewise, this disadvantage could be severe in most of the transfer-based fabrication techniques, which require the as-patterned nanostructures to be located on the target substrates. Specifically, the previous approach used an FeCl_3_ solution as the etchant, which is not compatible to other plasmonic materials such as Ag and Al.

Here, we report a fabrication technique of the template transfer method that utilizes a soluble adhesive, which is capable of the fabrication of freestanding nanomembranes and long suspended nanowires. The created nanomembrane which has a ~100 nm thickness features a periodic nanohole array of high quality and uniformity. For the proper use of these plasmonic structures, it is necessary to understand the physics behind their spectral properties. The previous theoretical analysis mainly focused on comparing the difference of the resonances between the freestanding and on-substrate cases. The freestanding nanomembrane shows symmetric bright and asymmetric dark resonance modes during transmission. In this work, it is further confirmed the antisymmetric mode is not really dark, but it has a prominent feature of optical reflection. As a versatile platform, these nanomembranes with nanohole arrays can also be used for the inverse fabrication of vertical Si nanowire arrays by the metal-assisted chemical etching method.

## 2. Materials and Methods

### 2.1. Design and Preparation

We have developed a template transfer process that is capable of the fabrication of metal nanostructures on unusual substrates [24]. In the previous method, once the metal structures have been transferred to the target substrate, they would permanently stay on the cured adhesive layer of insoluble epoxy. Along this technique route, a soluble adhesive is often used to release the as-patterned metal nanostructures from the templates without permanent attachment. Accordingly, we choose a water-soluble polymer, polyvinylpyrrolidone (PVP, CAS number: 9003-39-8) instead of an epoxy to temporarily hold the structures before releasing. In addition, the PVP adhesive has to be thick enough to avoid it flowing into the gaps between the nanostructures and touching the template. We note that some of the physical and chemical properties of the PVP have to be considered in the process design. For example, the melting point of the PVP is usually between 150 and 180 °C, so it is necessary to limit the ambient temperature to a proper range when it is combined with other applications. Since the PVP is incompatible with strong oxidizing agents, the following process may not fit into the fabrication procedures in the silicon industry and thus, specific measures should be taken during particular applications.

Moreover, a special holder is required to support the margin of the released metal nanostructures in order to make them freestanding. Firstly, the holder should consist of a flexible support to make conformal contact with metal membranes and a hard base to avoid the deformation of the entire frame during handling. Meanwhile, the surface of the support that contacts with the membrane has to be smooth, otherwise, even a small amount of roughness would result in wrinkles, which could extend from the interface to the other part of the nanomembranes. In addition, the inner edge of the support has to be round to form a smooth line of contact to avoid localized strain.

Based on these considerations, poly(dimethylsiloxane) (PDMS) (Sylgard 184, Dow Corning) and a Petri dish with a removable glass bottom are good choices for the holder preparation. Firstly, a precisely machined round steel rod with diameter of 5 mm was positioned perpendicularly on the glass bottom of the Petri dish. The rod’s diameter determines the size the freestanding part of the nanostructures. Thereafter, PDMS was cast into the Petri dish and cured at 75 °C for 2 h. After pulling the steel rod out of PDMS, we obtained a membrane holder with a complete hole in the PDMS on the bottom of Petri dish. The glass bottom was levered away from the Petri dish immediately before attaching the nanostructures.

### 2.2. Fabrication Procedure for Freestanding Nanostructures

The fabrication procedure of the freestanding metal nanostructures is shown in Figure 1a. A Si template that was patterned with the desired nanohole array was first deposited with 100 nm Au using e-beam deposition to form the metal nanomembrane. Then, the water solution of PVP (PVP40, Sigma-Aldrich, St. Louis, MO, USA, 40% in weight) was dropped on top of the template with the nanohole membrane. After drying, the PVP adhesive along with the Au membrane was then separated easily from the template using a razor. We note that only the very edge of the PVP/Au membrane was directly cleaved. The other area were consequently levered from the template due to the low adhesion of the Au membrane to the Si template.

Immediately after this, the Au membrane was attached to the previously prepared PDMS holder such that the membrane completely covered the hole in the holder. As the surfaces of the metal and PDMS were both very smooth, they were in conformal contact with each other. The van der Waals force between the metal and PDMS was strong enough to firmly hold the metal nanostructures without them detaching during the following steps and the test. The holder was then floated on water with the PVP adhesive being held under the water. After the bulk PVP was gradually dissolved, the holder with the metal nanomembrane was carefully transferred to new pure water sample to completely wash away the possible residue. The entire device was then taken out of water after 5 min. The metal membrane covering the hole of the holder became freestanding after the water evaporated.

The same procedure can be applied to make other freestanding structures such as nanowires. The Au nanowires were initially formed on the pattern surface of the Si template with the corresponding nanoslit array by e-beam deposition. Figure 1b highlights a 100-nm-thick freestanding Au nanomembrane and a nanowire array, which are suspended on PDMS holders. The nanomembrane has an area of about 25 mm^2^ with a hexagonal array of nanoholes. Different colours can be observed from both of the arrays due to strong diffraction from their periodic structures.

### 2.3. FDTD Simulation

Three-dimensional finite-difference time-domain (FDTD) simulations were performed using a commercial software, FDTD Solutions (Lumerical Inc., Vancouver, BC, Canada). A uniform mesh with a size of 2 nm (in the x, y, and z directions) was used. We set perfectly matched layer boundary conditions for the z direction, and periodic boundary conditions for x and y directions of the simulation region. The background refractive index for the freestanding nanomembranes was set to be 1 while it was in the air.

### 2.4. Metal-Assisted Chemical Etching Using Nanomembranes

The metal-assisted chemical etching (MaCE) was performed by immersing the Si wafer with the transferred nanohole membrane into an etching solution of 46 wt% HF (695068, Sigma-Aldrich) and 35 wt% H_2_O_2_ (1.08600, Sigma-Aldrich) (V_HF_/V_H2O2_ = 2:1). We note that the nanomembrane needs to be completely dried, and so we placed the Si wafer on the hotplate before the etching. The established conformal contact between the nanomembrane and Si surface prevented their separation during etching in the solution. By using electrons that were withdrawn through the interface between the metal and Si, the metal worked as an etching catalyst for the reduction of the oxidant (i.e., H_2_O_2_ + 2H+ + 2e− →2H_2_O), leading to the injection of positive holes (h+) into the valence band of Si. This enabled the oxidative dissolution of Si by HF at the interface between the metal and Si. Finally, the interplay between the oxidation of Si and the elimination of the oxidized silicon atoms by HF resulted in the continuous forward movement of the Au–Si interface. In the case of our nanohole array membranes which were used as the etching catalyst, the nanowire arrays can be formed as the complementary structure.

## 3. Results and Discussion

The freestanding metal nanostructures were characterized with SEM to confirm that they were intact and that they were the same as that on the template. Figure 2a demonstrates a freestanding nanomembrane of a hexagonal nanohole array with a periodicity of 700 nm and a diameter of 200 nm. The freestanding membrane has the smooth surface and the same well-ordered pattern as that of the template. To obtain information about the in-hole morphology, we cut off a piece from the membrane using focused ion beam (FIB) milling. Figure 2b displays the exposed cross-section of the nanomembrane, which reveals that the complete holes were open on both of the surfaces. Besides SEM imaging, we also used atomic force microscopy (AFM) to characterize the roughness of the Au freestanding nanomembrane. We obtain the AFM images at different regions of the membrane to make sure that it was intact and smooth. These images also confirm there was no bulk residue of PVP on the surface of these membranes.

Except for the rough edge, there was no other damage on the membrane as it was firmly held by the hard PVP during the releasing process. If only the central area of Si template was patterned, the periphery of the membrane would be a flat film without nanoholes after the Au deposition. Thus, the dried PVP edge would sit on this un-patterned area, whereby the entire patterned structures could keep intact after releasing. This allows for its successful bonding with the other structures or its incorporation in sandwich designs in possible applications.

The dimension of the freestanding membranes is limited by the template size and the solidified PVP on it. Since the Si template with a cm-size pattern was used in this work, the achievable size of freestanding membranes is in 1 cm range. Even larger membranes can be obtained by using a larger template. Accordingly, the use of a thicker PVP is necessary to firmly hold the larger membranes during the levering using a razor. Thus, it would take longer time to completely dissolve the bulk PVP by water. Using the setup in Figure 1b, a membrane collapse has hardly taken place during its lifting from water. However, it occurred easily when the size was within the wafer scale [24]. Such a failure is attributed to the uneven water distribution on the membrane surface that could induce the local strain. Moreover, once the membrane thickness is further reduced to about 50 nm, it becomes extreme fragile and can only float on the water surface. This strength issue also limits their application in the fields which require ultrathin membranes such as cryoEM sample supports. One possible way to avoid their failure is to place them on certain grids [15].

Nanowires can maintain their freestanding nature in the length scale of several millimeters as shown in Figure 2c. However, the collapse is predicted to easily occur if their lengths are more than one centimeter due to the surface tension of the water and their high aspect ratio. Most nanowires are constantly parallel as those on the template, while they attach onto each other as a result of the water drying (Figure 2c). We note that only the nanowires on the top relief of the template are released to be freestanding, unlike than those that are located on the bottom. Thus, this tight arrangement leads to dislocations and gaps at every other tens of nanowires. As shown in Figure 2d, one or several nanowires can also be suspended across the gap, and they are not parallel to the other nanowires any more once their either ends attach to different nearby nanowires. The width of the crevices between the nanowires is estimated to reach down to the sub-10 nm range. This could largely suppress the radiative scattering losses of these plasmonic structures as a result of the minimized scatter size when they are used as plasmonic sensors.

For their device applications, it is a prerequisite to realize the plasmonic mechanism of the freestanding membranes. The transmission and reflection spectra were measured at the normal incidence of the same freestanding nanohole array membrane (as shown in Figure 3a), respectively. Both of the spectra are normalized according to the peak at 500 nm for comparison. We can report that the resonant transmission efficiency significantly increases in the freestanding nanohole array membrane when it is compared to that on the substrate [24]. In the substrate case, the dielectric materials with different dielectric constants on either side of the membrane enable the resonances to occur at different wavelengths. In contrast, the uniform dielectric environment in the freestanding case matches the plasmon energy on both of the surfaces, resulting in the coincidence of the resonance on both of the surfaces and a remarkable transmission enhancement. This conclusion is obvious for the symmetric coupling mode, which shows itself as a transmission peak at 705 nm in Figure 3a. The field distributions of this mode in Figure 3c,e confirm this analysis. For the antisymmetric mode at 635 nm, its field distributions (Figure 3b,d) also show an enhancement. However, only a hardly observable peak appears at this resonance wavelength in the transmission. In contrast, the reflection of the freestanding nanohole array shows a notable valley at this resonance mode, corresponding to the missing enhancement character.

We attribute the different behaviors of the two coupling modes to their different loss mechanisms. In the symmetric mode at 705 nm, most of the field energy is concentrated around the edges of the nanoholes, which strongly scatter the optical energy, leading to a bright plasmon mode. This bright mode manifests itself as a peak during the transmission and a valley during the reflection. In contrast, the antisymmetric mode at 635 nm resonates above the flat Au surface and exponentially decays along the normal direction, whereby the nanoholes only scatter the incident light weakly to exit the other ends. This is the reason that the antisymmetric mode is dark during the transmission. However, the weak scattering results in a long resonance lifetime, leading to strong absorption by the metal. As a result, this absorption-dominating process presents an obvious minimum during the reflection when it is compared to that of a flat Au film (Figure 3a). In the previous analysis, only the symmetric mode that is dominated by scattering was considered in so-called extraordinary optical transmission [25]. The above analysis indicates that the antisymmetric mode is not a real dark mode in our freestanding case, but it has an obvious spectral feature of a 20% optical reflection decrease. In addition, the independence of absorption and scattering in the freestanding configuration could be promising for applications that separate the absorption and scattering processes. For example, the freestanding nanomembranes have the potential to realize the combination of particle manipulation and optical sensing: one wavelength can be used for plasmonic trapping and the other one can be used for fluorescent or Raman spectroscopy [26].

The freestanding nanomembranes can be transferred to various target substrates to be used as masks for nanostructure fabrication. To show this potential, the freestanding Au nanomembrane was utilized as an etching catalyst for the preparation of silicon nanowire arrays via MaCE (described in Section 2.4) as shown in Figure 4a. Instead of placing it in a PDMS holder, the nanohole membrane was released on the water surface after the dissolution of the PVP. The surface tension of water kept the floating membrane stretched, which was then picked up by using a Si (001) wafer. The Si under the metal catalyst was etched much faster than Si was without the metal coverage during the chemical etching. The morphologies of the resulting nanowires are defined by the parameters of the nanohole arrays, such as their diameter and period. The height of the array can be controlled by the etching time. The typical lengths of the Si nanowires in Figure 4b,c and d are around 1 μm, 2 μm, and more than 3 μm, respectively. The Si nanowires can stand on their own if their aspect ratios (length:diameter) are around below 10:1, otherwise they will bend in random directions due to surface tension during drying.

Although the anisotropic dry etching (e.g., DRIE) is also capable of realizing nanowires through masks that are made by either standard lithography or self-assembly, the pattern has to be generated each time for such in situ etching to be performed. Our method takes advantage of the template transfer and MaCE. First of all, the template can be re-used without repeated pattern fabrication. This replication technique is particularly suitable for the batch production of Si nanowires in applications such as energy harvesting and storage. Moreover, the MaCE itself is also a simple and low-cost method for producing diverse Si nanostructures with the ability to control various parameters [27].

## 4. Conclusions

In summary, we developed a fabrication scheme for freestanding nanomembranes and long suspended nanowires based on the template transfer method that utilizes a soluble adhesive. These nanostructures feature the same patterns with as high quality and uniformity properties as those of the corresponding template. The freestanding nanomembrane shows a symmetric and antisymmetric modes due to different hybrid coupling mechanisms of the plasmonic resonances from its two surfaces. Different spectral behaviors during transmission and during reflection are attributed to the different optical loss channels between the bright and dark modes. These nanomembranes with nanohole arrays can also be used for the inverse fabrication of vertical Si nanowire arrays by the metal-assisted chemical etching method. This technique shows promise for it to be further scaled up and to expand the capability of freestanding nanostructure array shaping for more photonic and plasmonic applications.

## Figures and Tables

**Figure 1 nanomaterials-12-03988-f001:**
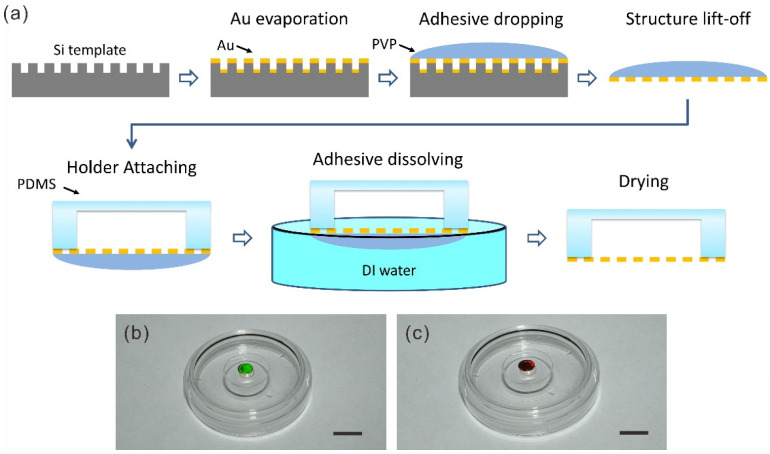
(**a**) Schematic of fabrication process of freestanding metal nanostructures with PVP adhesive: Au deposition, dropping PVP adhesive on the nanostructures, adhesive lift-off, attaching holder, dissolving adhesive in water, and drying freestanding nanostructures. Freestanding Au nanomembrane (**b**) and nanowires (**c**) on the PDMS holder in a Petri dish. Scale bars: 1 cm in (**b**,**c**).

**Figure 2 nanomaterials-12-03988-f002:**
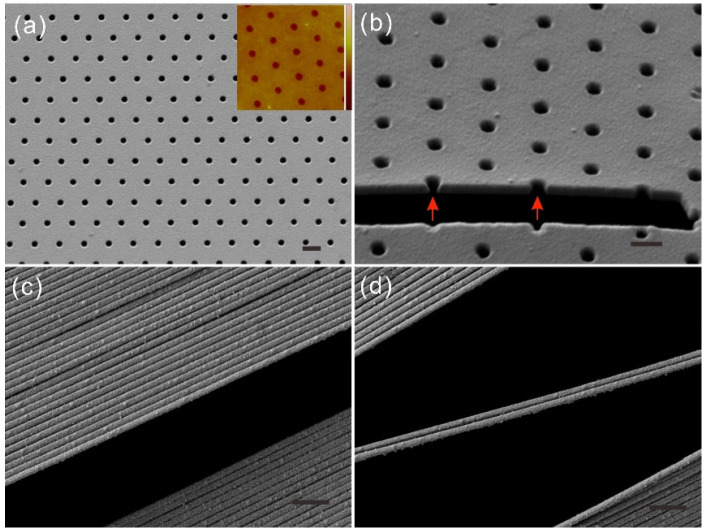
(**a**) Freestanding Au membrane with a hexagonal nanohole array with period of 700 nm and diameter of 200 nm. (Inset, AFM measurement; Color bar: 0 to 20 nm). (**b**) A cross-section of freestanding Au nanomembrane cut by FIB milling. Two arrows indicate the complete holes. Scale bars: 500 nm in (**a**,**b**). (**c**) Parallel freestanding nanowires with large gaps and sub-10 nm crevices. (**d**) Two dislocated nanowires suspending across the gap. Scale bars: 1 μm in (**c**,**d**).

**Figure 3 nanomaterials-12-03988-f003:**
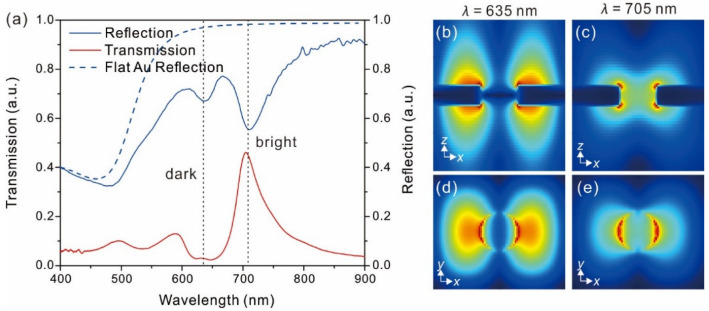
(**a**) Transmission and reflection of a freestanding membrane patterned with a hexagonal nanohole array which is compared with reflection of a flat Au film. Simulated electric field distributions at peak wavelengths of 635 nm and 705 nm in the x–z plane (**b**,**c**, respectively) and at the top air/Au interface (**d**,**e**, respectively). The array periodicity and hole diameter in the simulation are 700 nm and 200 nm, respectively.

**Figure 4 nanomaterials-12-03988-f004:**
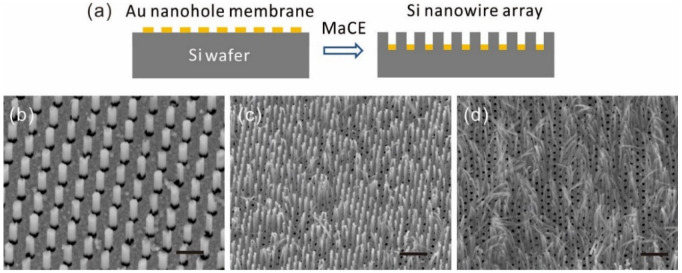
(**a**) Schematic of metal-assisted chemical etching with freestanding nanohole membrane for fabrication of nanowire array. (**b**–**d**) Nanowire arrays etched by using the same nanomembrane with different etching time of 1 min, 3 min, and 5 min, respectively. Scale bars: 1 μm in (**b**); 2 μm in (**c**,**d**).

## Data Availability

Not applicable.

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
