# Peer review of "Freestanding Metal Nanomembranes and Nanowires by Template Transfer with a Soluble Adhesive"

_nanomaterials, 2022, doi:10.3390/nano12223988_

Round 1

Reviewer 1 Report

This study deals with the transfer process of nanopatterns mediated by water-soluble polymers. However, I oppose the publication of this paper in terms of novelty.

Which polymer was used as the water-soluble adhesive in this study: PVP or PVA? In Figure 1(a), it is denoted as PVA.

Anyway, many papers have already been reported using PVP or PVA a water-soluble sacrificial layer for transfer printing (ex. https://doi.org/10.1038/s41598-020-61536-8).

The authors may have applied the above strategy to their previous paper, but I cannot agree that this is a challenging endeavor.

The application they present in Figure 4 is also not persuasive. As a silicon etching mask, a polymer pattern through photolithography is much more intuitive and cheaper. There is no need to use their complicated transfer process developed in this study and expensive gold mesh.

Reviewer 2 Report

The authors present the use of a water-soluble adhesive to transfer nanomembranes and nanowires between different surfaces. Given the relevance of nanostructures and the toxicity or environmental effect of common solvents, I can see an application for their technology, which makes it of interest to future readers. Unfortunately, except for the introduction, the manuscript’s language requires substantial improvements and clarifications, as, in the current state, numerous arguments brought forward by the authors are difficult to understand and evaluate. Additionally, I have several major and minor concerns regarding the novelty and relevance of the presented work, which I would like the authors to discuss before making a suggestion concerning a possible publication of their manuscript:

Major:

The authors state that the gold membrane is simply detached from the Si template using a razor. However, while it seems quite challenging, could that lead to damage of the structure (which, even if just locally, could be crucial for subsequent applications)? And if that is feasible, why would their method with the adhesive even be required anymore as it might be possible to simply pick it up the membrane directly with the PDMS as a carrier using a razor blade? Relatedly, if alternative ways for detachment would be considered, how is the temperature stability of the water-soluble adhesive? And, given that chemical processes might still be required to lift off the membrane from the template, are there limitations in terms of the adhesive’s chemical stability for prominent chemical etchants used in Si industry?

While the spectral analysis and resonance simulation seem to be well done, it is unclear why this is of relevance for the presented work. It is correct that freestanding nanohole arrays react differently than membranes on a flat substrate. However, as the authors also state themselves, this has been shown previously in one of their own publications and, hence, does not add anything to the topic of the presented work, which is “the introduction and testing of a transfer approach for nanomembranes and nanowires using a water-soluble adhesive”.

Given that it is not clear why the discussion concerning the resonance is of relevance to this work, it should be removed from the abstract as it substantially hinders the accessibility of the manuscript. Relatedly, it is not clear why this topic is discussed in detail at the end of the introduction before even presenting the corresponding results or highlighting the relevance of this investigation in terms of “membrane transfer with water-soluble adhesive”.

The authors repeatedly state that their method can be used for “large-area” membranes. Hence, it is relevant to discuss possible limitations or maximum dimensions. What is the maximum size that can still be transferred with this technique without damaging the potentially valuable membranes?

Given that the topic of the manuscript focuses on testing new materials for complex procedures, the authors should provide the exact specifics of all used materials, e.g., component numbers and manufacturers.

Line 50 as well as twice in conclusion: “Green fabrication technique” is misleading as the initially required methods to get all the nanowires and the membranes still require regular etching techniques and silicon processing. Hence, given that the work focuses only on this type of application, the statement is misleading even if PVP itself is not known to be environmentally hazardous.

Line 81 and following: Why do the authors rely on a 5 mm wide steel rod to produce the mould for the PDMS rather than regular SU-8-based lithography to allow for arbitrary shapes that might even improve final alignments? Especially given the resulting strong variation in dimension once the rod is pulled out of the PDMS. Relatedly, are there any limitations in terms of height or aspect ratio for the holder (PDMS) to prevent surface forces induced by the dissolution of the adhesive from collapsing the thin membranes? Finally, it is not clear what the difference between the support and the holder is. Please at these labels to Figure 1.

Line 100 – 103: Did the authors experience issues with samples lost in water due to a lack of adhesion between the sample and the PDMS? One would expect that this effect is even more enhanced due to the strong surface tension of the water and since gold is hydrophilic while PDMS is hydrophobic. Relatedly, did the authors experience any collapse of nanomembranes or wires due to surface tensions during water evaporation? Especially since this is a common issue in silicon processing, which is why many applications rely on the use of critical point dryers.

Line 209: In contrast to the statement made by the authors, the stability has likely nothing to do with the etch duration but is rather based on the resulting aspect ratio. Hence, please state what the aspect ratios for the silicon nanowires are. Additionally, it is unclear how this application should show why this work is important. While it is correct that nanomembranes can be transferred to a Si wafer such that nanowires can be etched later on, given that the nanomembranes anyway first rely on Si processing and, hence, the fabrication of a mask on a silicon wafer, the overall process of actually transferring the membrane seems irrelevant.

Minor:

Abstract: State clearly what materials the nanomembrane and the nanowires are made of as well as what adhesive is used, such that future readers can evaluate the suitability of the presented methods as easily as possible. Relatedly, I would advise against the use of “without chemical process” but clearly state that a water-soluble adhesive is used to prevent chemical etching (not chemical process) throughout the manuscript to prevent misconceptions.

Figure 1: Possibly due to a lack of care during preparation, but is it now PVA (as labelled in the figure) or PVP (as written in the manuscript)?

Figure 1: Please add a schematic to show how the transfer of suspended nanowires is performed (including a discussion concerning the challenges of adding the water-soluble PVP to already suspended wires).

Figure 2: It is not clear what Figure 2b really shows as the holes are, at least on my computer screen, only visible on the surface but not throughout the material (in contrast to what one would usually expect from a side view resulting from FIB).

Figure 2a: What is the height of the colour bar shown for the AFM? Unfortunately, it lacks all labels.

Figure 3: What is the size of the scale bars?

Reviewer 3 Report

This manuscript developed a green fabrication scheme for large-area freestanding nanomembranes based on the template transfer method that utilizes a soluble adhesive. By using the nanomembranes as an etching catalyst, long Si nanowires were fabricated. The fabrication method is quite novel and the manuscript is well organized. However, the analysis of the plasmonic resonances sounds not that professional. So I suggest that this paper is suitable for publication in Nanomaterials after the major revision:

1.       In Introduction Lines 34-37, references are needed for the membrane applications.

2.       In Introduction Lines 38-39, the reference is not published by the authors but they claimed that the membranes were demonstrated by them. Maybe revise it as “For example, the freestanding nanohole array membranes we recently demonstrated [cite reference 22] can work as sample supports in single-particle cryoelectron microscopy with sub-1 Å specimen movement [13].”

3.       In Page 5 Lines 184-199, it seems that there is localized plasmon resonance in the nanoholes at 635 nm leading to low reflection and transmission. The analysis here sounds very strange. Is there any reference using dark and bright (symmetric and antisymmetric) modes to describe the plasmonic resonances in the metal membranes?

4.       For Figure 4, the value of the scale bar is not mentioned. Also what is the typical length of the Si nanowires in Figure 4 b, c and d, respectively?

Round 2

Reviewer 2 Report

I acknowledge the authors' adjustments to the manuscript and the provided direct responses. While the manuscript has certainly improved with regard to overall accessibility (e.g. by simplifying the use of “holder” and “support”), I fear that the authors have still not been able to properly assess all of my major concerns – a factor which I believe would be crucial for a publication in Nanomaterials, even more so given the overall expectations set by its journal metrics. Hence, I would like to provide the authors with an additional chance to properly highlight the relevance of their work and the corresponding scientific novelty before making a final recommendation concerning publication. Furthermore, I would invite the authors to include further details as provided in the direct responses in the manuscript, such that future readers would be aware of the challenges and limitations of the presented technique.

Major:

The authors acknowledge that the application of a razor blade leads to damage at the edge of the membrane. Unfortunately, nothing related to this limitation or any other restriction has been added to the manuscript. However, I would suggest the authors discuss possible applications where such damage would not lead to possible issues, especially since most devices that would rely on pre-fabricated membranes would certainly require them to be purely flat to, e.g., allow for the successful bonding with other structure and incorporation in sandwich designs.

Relatedly, the presented application of the gold membrane for subsequent etching of micropillars, seems rather irrelevant as this task can already easily be performed without the tedious transport of a gold membrane in the first place using, depending on the dimensions, either standard lithography or self-assembled etching masks (with down to 35 nm pore sizes that have been used by the reviewer in combination with high-resolution DRIE personally despite the authors’ false claim that such feature sizes are not possible with other methods).

While it is correct that, following fabrication, proper characterisation of the material is crucial, I still strongly disagree with the current presentation of the results based on spectral analysis as, given that it is solely an evaluation method and not the key topic of the manuscript (at least according to the title), the way it is currently highlighted and discussed significantly hinders the general accessibility of the manuscript’s main work. Even more so as, despite its apparent novelty and major importance to the overall publication, a proper discussion of the results is lacking. Furthermore, the use of non-scientific terminologies like “extraordinary reflection” without corresponding evaluation does not improve this case.

Process parameters, including the high 150 – 180 °C melting point of PVP which the authors used as part of their protocol and mentioned in the direct response but not in the manuscript, are absolutely crucial and have to be mentioned and discussed in detail to allow for proper evaluation of their methods for other applications. Furthermore, while the authors argue that the chemical stability of PVP is not relevant as it is only in contact with gold and water, this should still be further discussed as it is relevant to see how it would fit into more complex fabrication procedures required in silicon industry.

Minor:

While the authors removed their statement on “large scale” applications, it would still be relevant that they clearly state what has been done and where possible limitations with regard to size might be.

I’m having trouble following their response concerning the possible collapse of the structures during transport: What do the authors mean that they “haven’t experienced the detachment or collapse of nanowires in the length scale of several millimeters”? Is this referring to large-scale collapses or is this referring to no collapses at all even (not even local folding due to surface forces) for samples that are several millimetres in dimension?

What is the hole diameter used for the simulation presented in Figure 3?

Reviewer 3 Report

The authors have answered my questions. So I suggest that the revised manuscript is suitable for publication in Nanomaterials.

Author Response

Thanks for your review and help. 

Round 3

Reviewer 2 Report

The authors have successfully addressed my major concerns through the repeated revision of the manuscript. Specifically, the extended discussion regarding the relevance of the template transfer and the reuse of the template for etching procedures ensures that the proposed method is accompanied by a meaningful technological application. Furthermore, through the general improvement of the manuscript via the inclusion of information that was previously only provided as part of the direct responses, I believe the authors were able to ensure that their work is now accessible to a broader audience. While I would still invite the authors to extend the discussion of possible limitations of their approach (not only on the material side), the only other minor inputs are as follows:

Please check the manuscript in detail to prevent possible misconceptions and improve overall language use (including newly added sections).

Please add scale bars to 1b+c.

Author Response

Thanks for the reviewer's advice. We have thoroughly checked the manuscript and made some minor changes. Also, the scale bars have been added to Figure 1 b and c with the caption. The possible limitation of our method is the membrane cannot be too much thin to be picked up from water surface. Beyond, we have also discussed the strenth issue in practice applications and added a note: "Moreover, once the membrane thickness is further reduced to about 50 nm, it becomes extreme fragile and can only float on the water surface. This strength issue also limits their application in the fields that require ultrathin membranes such as cryoEM sample supports. One possible way to avoid their failure is to place them on certain grids [15]."